# Effects of Repetitive Transcranial Magnetic Stimulation (rTMS) Combined with Aerobic Exercise on the Recovery of Motor Function in Ischemic Stroke Rat Model

**DOI:** 10.3390/brainsci10030186

**Published:** 2020-03-23

**Authors:** Juanxiu Cui, Cuk-Seong Kim, Yeongwook Kim, Min Kyun Sohn, Sungju Jee

**Affiliations:** 1Department of Rehabilitation Medicine, School of Medicine, Chungnam National University, Daejeon 35015, Korea; cjx858231@gmail.com; 2Department of Physiology & Medical Science, Chungnam National University College of Medicine, Daejeon 35015, Korea; 3Department of Rehabilitation Medicine, Chungnam National University Hospital, Daejeon 35015, Korea

**Keywords:** transcranial magnetic stimulation, exercise, neuroplasticity, function recovery, middle cerebral artery occlusion

## Abstract

The therapeutic benefits of repetitive transcranial magnetic stimulation (rTMS) combined with rehabilitation therapy on recovery after stroke have not been fully elucidated. This study aimed to explore the therapeutic effects of rTMS followed by aerobic exercise on neuroplasticity and recovery of motor function in a rat model of permanent middle cerebral artery occlusion (MCAO). Rats were randomized into sham operation (*N* = 10, sham op), MCAO (*N* = 10, control group), rTMS (*N* = 10, MCAO and rTMS therapy), and combination groups (*N* = 10, MCAO and combination therapy). High-frequency rTMS (10 Hz) was applied on the ipsilesional forepaw motor cortex, and aerobic exercise training on the rotarod was performed for two weeks. The rotarod and Garcia tests were conducted to evaluate changes in behavioral function. Motor evoked potentials (MEPs) were used to evaluate electrophysiological changes. Stroke severity was assessed using infarction volume measurement. Neuronal recovery was explored with western blot for brain-derived neurotrophic factor (BDNF) pathway proteins. Compared with control therapy, combination therapy was significantly more effective than rTMS therapy for improving function on the rotarod test (*p* = 0.08), Garcia test (*p* = 0.001), and MEP amplitude (*p* = 0.001) In conclusion, combination therapy may be a potential treatment to promote recovery of motor function and neuroplasticity in stroke patients.

## 1. Introduction

Ischemic stroke is the leading cause of severe and long-term physical disability in the United States [1]. Hemiparesis reduces the quality of life and causes a major economic burden to families and society. At present, rehabilitation therapy is the best approach for the treatment of neurological deficits after stroke [2]. However, neurological impairments persist in most stroke survivors after rehabilitation, highlighting the inadequacy of treatment. Thus, improving the effectiveness of stroke rehabilitation remains a critical unmet need.

Repetitive transcranial magnetic stimulation (rTMS) is a noninvasive neuromodulation technique. rTMS technology is applied to stroke patients in various ways such as high frequency, low frequency, and intermittent theta-burst stimulation (iTBS), which produce different regulatory effects. Recently, animal studies have reported that rTMS improved functional motor recovery, enhanced brain-derived neurotrophic factor (BDNF)/tropomyosin-related kinase B (TrkB) signaling [3,4], and was associated with anti-apoptotic mechanisms [4,5].

Exercise training is a simple and widely practiced behavior that is used in post-stroke rehabilitation to promote recovery of mobility deficits. Previous animal research using a middle cerebral artery occlusion (MCAO) model reported that exercise training improved motor behavior and altered the expression of BDNF and TrkB proteins, which are associated with plasticity, and promoted brain vascularization neurogenesis, functional changes in neuronal structure, and neuronal resistance to injury in the hippocampus [6,7,8,9,10,11]. Ding et al. [12] reported that motor balance and coordination training enhanced functional outcomes in rats with MCAO.

BDNF and its cognate receptor TrkB, a member of the neurotrophic receptor tyrosine kinase family, and AKT is one of the downstream signaling pathways to BDNF that activates TrkB, play important roles in neuronal survival, proliferation, maturation, outgrowth in the developing brain, and modulation of synaptic plasticity, which have recently been shown to be critical for functional recovery after stroke [13,14]. 

Most studies have investigated the mechanisms of functional recovery after either rTMS or exercise in isolation [3,13]. Our previous research demonstrated that high frequency rTMS led to a rapid recovery of behavioral performance compared with that following sham stimulation in rats with permanent MCAO [15]. Although rTMS or exercise in isolation improves motor function and enhances neuronal plasticity, to the best of our knowledge, the use of both approaches in combination to treat permanent MCAO has not been investigated to date. Indeed, the therapeutic benefits of rTMS combined with rehabilitation therapy on recovery after stroke have not been elucidated. Therefore, this study aimed to explore the therapeutic effects of rTMS and exercise on the recovery of motor function during the first two weeks after permanent cerebral ischemia in rats. We hypothesize that low-intensity aerobic exercise combined with high-frequency rTMS improves motor recovery and enhances neuroplastic changes after stroke compared with rTMS alone.

## 2. Materials and Methods

### 2.1. Animals

Male 8-week-old Sprague-Dawley rats (weighing 230–250 g) were used. Rats were purchased from a commercial experimental animal center and housed in the colony room under controlled temperature (22 °C) and 12-h light/dark cycle throughout the protocol with free access to food and water. Before experimentation, all rats were subjected to habituation periods for experimental tasks such as the rotarod task for three days [7,16]. With the exception of the sham operation group, other groups received MCAO. Rats were included for experimentation if they scored 8–12 points on the Garcia test used for evaluating motor impairment 72 hours after the operation, and were used for subsequent experiments. The rats were divided into the sham operation, control, rTMS, and combination groups by random allocation. The rTMS group received rTMS treatment only. The combination group received rTMS, followed by exercise treatment. The sham operation and control groups received sham stimulation. The experimental procedures (Figure 1) were approved by the Institutional Animal Care and Use Committee (IACUC) of the preclinical research institute at Chungnam National University Hospital (CHUH-016-A0012-1).

### 2.2. Permanent Middle Cerebral Artery Occlusion Model (MCAO)

Rats were anesthetized with 2,2,2-tribromoethanol (avertin; 200–240 mg/kg, intraperitoneal injection) using a modified Longa method [17] to generate a permanent MCAO rat model on the right side. First, the areas under the chin were sterilized with an alcohol swab, and a midline incision was made between the manubrium and jaw. Layers were sequentially stripped until Y-shaped artery bifurcations were exposed including the right central carotid artery (CCA), right external carotid artery (ECA), and right internal carotid artery (ICA). A loose collar suture was tied around the CCA. A tight collar suture was tied 1 cm from its bifurcation to the ECA. A microvascular clamp was applied to the ICA. A hole was pinned with a needle 0.4 cm from its bifurcation to the ECA. A filament (Doccol Corporation, Sharon, MA, USA) was then introduced into the hole approaching the CCA, and a loose collar suture was tightened around the filament. An arteriotomy was performed between the hole and the tight collar suture. The filament direction was altered toward the ICA and guided up the ICA until resistance was felt. The loose collar suture was then tightened around the inserted filament to prevent movement. Skin wounds were stitched. Rat body temperature was monitored throughout surgery, occlusion, and reperfusion with a rectal thermometer and maintained between 36.0 °C and 38.0 °C via a flat heat pad. After recovery from anesthesia, rats were returned to their home cages with free access to food and water. Rats in the sham operation group underwent a sham operation with the exception of the filament step.

### 2.3. Neuromodulation

A magnetic stimulator (MagPro X100, MagVenture, Inc., Alpharetta, GA, USA) with a figure-of-eight coil (cool-B35; 24 mm inner diameter, 47 mm outer diameter) was used to stimulate conscious rats three days after cerebral ischemia. The maximal initial dB/dt near the coil surface was 50 kT/s and the active pulse width (biphasic) was 290 μs, according to the magnetic and electrical properties of the sheet of cool-B35 coil supplied by the manufacturer. The stimulation site was the hot spot in the right cerebral hemisphere that was measured and marked before surgery. Selection criteria for the hot spot were adapted from previous reports [18]. The resting motor threshold (RMT) was defined as the minimum intensity that caused the motor-evoked potentials (MEP) to exceed 50 μV in at least half of the 10 trials. We found the intensity of RMT on the right hemisphere before the MCAO procedure. We conducted a pilot trial to determine the RMT for forepaw in 10 healthy rats before the main experiment. The pilot trial showed that the average RMT for biceps muscle was 37% (± 1.08) of machine output. The stimulation was performed at 10 Hz, 80% of RMT, 1-s stimulation, 6.4-s rest, 400 pulses per day, and 10 sessions a day for two weeks [3,5,19]. Stimulation was performed 4000 times in total.

Sham rTMS was performed 30 cm above the hot spot in the opposite coil direction with the same protocol (Figure 2).

### 2.4. Aerobic Exercise

In the combination group, rats participated in rotarod exercise immediately after the end of the neuromodulation session. Initially, rats were required to run at a speed of 5 and 15 revolutions per minute (rpm) for 20 min over 3–5 days, followed by 25 rpm thereafter over 20 min over subsequent days. If the rats were unwilling to exercise, they were encouraged by gently pushing their hips. The exercise protocol was based on previous studies, with some modifications [7,11,12] due to exercise equipment.

### 2.5. Garcia Test

The Garcia test consists of six items (scoring range): spontaneous activity (0–3); symmetry in four limb movement (0–3); forepaw outstretching (0–3); climbing (1–3); body proprioception (1–3); and response to vibrissae touch (1–3). The total score for each rat ranged from 3 (severe deficit) to 18 (no deficit) points. In this study, we used the total Garcia score.

### 2.6. Rotarod Test

The rod diameter was 9 cm, and rotation velocity was increased from 5 to 50 rpm, with an increase of 5 rpm every 30 s. Performance was measured three times and averaged. The latency (s) and distance (m) to the first fall were recorded until the rat fell from the rod or grasped the device, and were used for data analysis. The neurobehavioral outcomes including the rotarod test and Garcia test were measured at pre-operation and 3, 7, 10, and 15 days after operation.

### 2.7. Electrophysiological Evaluation

To record MEPs, an active needle electrode was placed on the biceps muscle belly. A reference needle electrode was placed between the second and third forepaw digits. A ground electrode was placed subcutaneously above the tail. Electromyography signal was acquired with settings according to motor nerve conduction methods in the electrodiagnostic machine (20 to 10,000 Hz band pass filter setting; Keypoint 4^®^, Dantec™, Skovlunde, Denmark). MEPs were collected with stimulation intensity at 120% of the initially predetermined RMT (37% machine output) at the hot spot with the figure-of-eight coil (Magstim Rapid2 Carmarthenshire, UK). MEPs were measured at pre-operation, 3, and 15 days after operation. To ensure stability of the recordings, all rats were placed under general anesthesia with an intraperitoneal injection of avertin. We used a maximal value of baseline to negative peak amplitude (mV) at each period. Changes in amplitude between 3 and 15 days were acquired for statistical analysis. MEP amplitude provides a measure of the change in excitability of polysynaptic neural pathways descending from the motor cortex to the target muscle. Changes in MEP amplitude are related to brain plasticity.

### 2.8. Histologic and Biochemical Evaluation

#### 2.8.1. Infarct Volume Measurement

Triphenyltetrazolium chloride (TTC) staining was used to measure the infarct volume. Each group of rats (*N* = 5) were sacrificed 15 days after the operation. Whole brains were removed and coronally sectioned at 2.0 mm thickness using a rat brain slicer (Zivic Instruments; Pittsburgh, PA, USA). Five brain slices from each rat were incubated in 2% (*w*/*v*) TTC solution for 20 min at 37 °C. After staining, the sections were photographed to determine the ischemic infarction volume. Infarct volume was measured using the NIH image program (National Institutes of Health, Bethesda, MD, USA) analysis. The infarction area was calculated by dividing the sum of infarction areas of each of the five slices by the whole brain area of each section.

#### 2.8.2. Western Blotting

Western blotting was performed to detect protein expression of BDNF, TrkB, P-TrkB, AKT, and P-AKT in the peri-infarct region of the ipsilateral brain cortex at 15 days (*N* = 5 per group). Under deep anesthesia, rats were sacrificed, and brain tissue was removed and dissected. Cortical tissue was homogenized in lysis buffer, approximately 15–20 μg of protein was loaded on 9–18% gels and transferred to membranes. The membranes were incubated with anti-BDNF (1:1000; Abcam), anti-total-TrkB (1:1000, Cell Signaling Technologies), anti-phosphorylated-TrkB (1:1000, Cell Signaling Technologies), anti-total-AKT (1:1000, Cell Signaling technologies), anti-phosphorylated–AKT (1:1000, Cell Signaling Technologies), and anti-β-actin (1:1000, Cell Signaling technologies) antibodies overnight at 4 °C. The membranes were washed with Tris-buffered saline with 0.1% Tween 20 (TBST) and incubated with appropriate secondary antibodies for 1 h. After washing with TBST, immunoreactivity was visualized by using the enhanced chemiluminescence method. The bands were quantitated using the NIH image program (ImageJ) analysis.

## 3. Statistics

Windows SPSS 25.0 (version 25.0; SPSS, Chicago, IL, USA) was employed for statistical analysis. All data were presented as mean ± standard error of the mean (SEM). Repeated-measures analysis of variance (RMANOVA) was used to explore the effect of time and interaction of time × intervention on the electrophysiological and behavioral data. As some data were not normally distributed, non-parametric statistical analyses were used. The Kruskal–Wallis test was used to compare the differences among the three groups. If a significant difference was detected, post-hoc analysis was performed using the Mann–Whitney U-test with Bonferroni correction. The statistical significance levels for RMANOVA and post-hoc analysis were set at *p* < 0.05 and *p* < 0.017, respectively. Spearman’s correlation analyses were carried out to the analyze the correlation between the increases in BDNF and the change in MEP amplitude.

## 4. Results

### 4.1. Animals

In total, 48 rats underwent MCAO surgery with an average operation time of 12.65 min. Of these, five rats died within three days post-operation, and three rats failed the Garcia score. The surviving rats were randomly assigned to three groups. Over the course of treatment, eight rats died across the three groups. The final numbers of animals in the control, rTMS therapy, and combination therapy groups were 10, 10, and 12, respectively.

### 4.2. Behavior Test

#### 4.2.1. Rotarod Test

The third day after the operation, the values of latency to fall in the control, rTMS, and combination groups were 47.28 ± 4.22 s, 47.17 ± 5.79 s, and 56.84 ± 5.88 s, respectively. No significant differences were observed among the three groups (*p* > 0.05). With regard to latency to fall for the three MCAO groups, RMANOVA revealed a significant effect of time (F(3, 108) = 67, *p* < 0.001) and intervention × time interaction (F(6, 108) = 4.7, *p* < 0.001). Post-hoc analysis with the Mann–Whitney U test with Bonferroni adjustment revealed that the latency to fall in the combination group (122.07 ± 15.88 s) was significantly longer than that for the other two groups (control group: 63.5 ± 7.9 s; rTMS group: 65 ± 5.72 s) at day 7 post-operation (*p* < 0.017). The latency to fall was not significantly different between the rTMS and control groups (*p* > 0.017). On the tenth day, the mean latency to fall was significantly longer in the combination group (151.69 ± 16.18 s) than in the other two groups (control: 82.58 ± 10.5 s; rTMS: 94.35 ± 3.99 s; *p* < 0.017). The latency to fall was significantly longer in the rTMS group than in the control group (*p* < 0.017). On the fifteenth day, the mean latency to fall was significantly longer in the combination group (155.92 ± 15.64 s) than in the other two groups (control: 101.75 ± 7.89 s; rTMS: 105.21 ± 7.24 s; *p* < 0.017). The latency to fall was not significantly different between the rTMS and control groups (*p* < 0.017).

The distance to fall exhibited the same pattern as that of latency to fall. On the third day after the operation, the mean values of distance to fall in the control, rTMS, and combination groups were 1.37 ± 0.27 m, 1.36 ± 0.15 m, and 1.86 ± 0.27 m, respectively. No significant differences were observed among the three groups (*p* > 0.05). Regarding fall latency of the three MCAO groups, RMANOVA revealed a significant effect of time (F(3, 108) = 35.28, *p* < 0.017) and intervention × time interaction (F(6, 108) = 4.84, *p* < 0.017). On the seventh day, the distance to fall in the combination group (6.75 ± 1.63 m) was significantly longer than that of the two other groups (control: 2.2 ± 0.41 m; rTMS: 2.9 ± 0.37 m; *p* < 0.017). No significant differences were observed between the rTMS and control groups (*p* > 0.017). On the tenth day, the distance to fall was significantly longer in the combination group (9.45 ± 1.87 m) than in the other two groups (control: 3.42 ± 0.85 m; rTMS: 3.9 ± 0.28 m; *p* < 0.017). Distance to fall was significantly longer in the rTMS group than in the control group (*p* < 0.017). On the fifteenth day, the distance to fall was significantly longer in the combination group (9.85 ± 1.90 m) than in the two other groups (rTMS: 4.8 ± 0.62 m; control: 4.52 ± 0.62 m; *p* < 0.017). Distance to fall was significantly longer in the rTMS group than in the control group (*p* < 0.017). The temporal changes in latency and distance to fall in each group are shown in Figure 3.

#### 4.2.2. Garcia Test

On the third day post-operation, the values of the Garcia score in control, rTMS, and combination groups were 8.3 ± 0.3, 8.89 ± 0.45, and 9.6 ± 0.43, respectively (*p* > 0.05). The detailed results of the Garcia test are shown in Figure 4.

RMANOVA revealed a significant effect of time (F(2, 65.7) = 157, *p* < 0.001) and intervention × time interaction (F(6, 65.7) = 6.5, *p* < 0.001) on Garcia score. Post hoc analysis revealed that the combination group had significantly higher scores than those of the other two groups at 7 (control: 8.9 ± 0.34, *p* < 0.017; rTMS: 9.9 ± 0.43, *p* < 0.017; combination: 13.09 ± 0.37, *p* < 0.017), 10 (control: 10.3 ± 0.3, *p* < 0.017; rTMS: 12 ± 0.42, *p* < 0.017; combination: 13.64 ± 0.39, *p* < 0.017), and 15 (control: 12.5 ± 0.37, *p* < 0.017; rTMS: 13.5 ± 0.27, *p* < 0.017; combination: 14.82 ± 0.12, *p* < 0.017) days after operation. On day 7, the rTMS group and control group scores were not significantly different (*p* > 0.017). On days 10 and 15, the rTMS group score was higher than that of the control group (*p* < 0.017).

### 4.3. Electrophysiological Evaluation

The third day post-operation, MEP amplitudes in the control, rTMS, and combination groups were 0.44 ± 0.04 mV, 0.47 ± 0.64 mV, and 0.53 ± 0.047 mV, respectively (*p* > 0.05). MEP was measured on days 3 and 15 after MCAO. The results are shown in Figure 5. MEP amplitude changes between days 3 and 15 in the combination group (0.186 ± 0.295 mV) were significantly larger than those in the control group (0.057 ± 0.029 mV; *p* < 0.017) but not those in the rTMS group (0.157 ± 0.02 mV). No significant differences were observed between the two intervention groups.

### 4.4. Histologic and Biochemical Evaluation

#### 4.4.1. Infarct Volume

TTC staining was performed on day 15. Results are shown in Figure 6. The infarction area in the control, rTMS, and combination groups were 22.72 ± 2.87, 20.70 ± 1.63, and 18.84 ± 2.49%, respectively. There were no significant differences among the three groups (*p* > 0.05).

#### 4.4.2. Western Blotting

Cortical tissues in the ipsilateral hemisphere excluding the infarct area were used for western blotting. The protein levels of BDNF, P-TrkB, T-TrkB, P-AKT, and T-AKT are shown in Figure 7. A Kruskal–Wallis test revealed higher protein levels of BDNF, p-TrkB, and p-AKT in the combination group compared to that in the control group (*p* < 0.017), but not in the rTMS group. No significant differences were observed between the two intervention groups. Spearman’s test revealed significant correlation was observed between increases in BDNF and change in MEP amplitude (*r* = 0.598, *p* < 0.05).

## 5. Discussion

To the best of our knowledge, this is the first report on the effects of combined rTMS and aerobic exercise on motor recovery in a rat model of permanent stroke. This study demonstrated that the combination of neuromodulation and exercise resulted in greater improvements in behavioral and neurophysiologic tests compared to that in the control group. Based on rotarod and Garcia scores, there was greater improvement in the combination group than in the rTMS group, although there were no significant differences in MEP and infarct volume between the rTMS and combination groups. Moreover, our results revealed that the protein levels of BDNF and p-TrkB were upregulated in the combination group relative to those in the control groups. In accordance with these results, motor function and protein expression levels of BDNF and TrkB activation were significantly increased by combined rTMS and aerobic exercise, indicating that this approach may be a promising strategy for stroke rehabilitation.

The combination of rTMS and aerobic exercise has been assessed in several studies to examine the potential benefits on motor function in several diseases. Previous research reported that combined therapy was feasible and could improve depressive symptoms and walking capacity [20]. Wang et al. reported that the combination of rTMS and treadmill training enhanced the effect of treadmill training on the modulation of corticomotor inhibition and improvement in walking performance in Parkinson’s disease patients [21]. Studies using individual treatment have reported that high-frequency rTMS [3,5,22] and exercise training [8,9,13,23,24,25] exerted beneficial effects on functional recovery. The results of our study showed consistent findings with previous studies. Our study additionally showed that combination therapy was significantly more beneficial than rTMS in isolation. Lan et al. [13] reported that treadmill exercise therapy caused a significant improvement in 19 days compared to no intervention on modified neurological severity score. Notably, we observed that compared to the control group, the combination group exhibited significant improvement in seven days. In addition, we observed that the combined group showed significantly greater improvements than those of the other two groups at every time point in the rotarod test. Although the Garcia score improved, it was not consistent with the rotarod test results, and differences among the three groups compared with those for the rotarod test were relatively small. We assume that the difference between these two behavioral tests reflects the effects of repetitive exercise. The combined group used the rotarod every day, whereas the Garcia test does not reflect a training effect. We thus speculate that the Garcia test solely reflects the effect of rTMS.

In our study, MEP amplitude was significantly improved in the combination group compared to that in the control group, but there were no additional improvements beyond those exhibited by the rTMS group. Furthermore, no significant differences were observed between the rTMS and control groups. MEP amplitude after rTMS combined with a range of motion exercise was significantly increased when compared to that after no therapy or rTMS treatment in patients after stroke [26]. Our findings are consistent with previous studies. High-intensity training improves aerobic fitness and grip strength, and promotes cerebral plasticity after focal ischemia [25]. We observed a trend for higher MEP amplitude in the rTMS group compared to that in the control group and lower MEP amplitude compared to that in the combination group. These findings are consistent with previous studies.

In our study, there were no significant differences among the three MCAO groups in infarction area. Previous research reported that relative infarct volume was reduced by high-frequency rTMS therapy [3] and exercise [22,27]. In our study, we used permanent MCAO models, but past studies [3,27] used the temporary model, which induced a relatively small size of infarction. We used TTC staining to measure the infarcted area at 15 days after stroke. The brain atrophy and necrosis could affect the measurement of infarction volume, even though we handled the brain very carefully. However, we used the same staining method for measurement before, and the present results are consistent with our previous studies [15].

The mechanisms of motor recovery include the expression of growth factors, axonal sprouting, synaptogenesis, and synaptic strengthening [21,22]. BDNF plays a role in neuronal survival, neural repair, and neural plasticity [14,28]. Previous studies have reported that exercise [6,13,24,25] or rTMS [3,29,30,31] activates the BDNF/ TrkB signaling pathway. Initially, our results demonstrated that the protein levels of BDNF, p-TrkB, and p-AKT were elevated to a greater degree in the combination group compared to that in the control group. Although the difference was not significant, the trend of protein expression in the rTMS group was higher than that of the control group, but lower than that of the combination group. We speculate that combination therapy improves neural plasticity, and the combination treatment was more effective than rTMS. We demonstrated elevated BDNF and TrkB protein expression in the combination group. To our knowledge, the effects of combination therapy on protein expression have not been reported. However, Zheng et al. [25] reported that forced exercise elevated serum corticosterone concentration and reduced BDNF levels in the rat brain. These differences may be due to the starting time and intensity of training. In Zheng et al.’s study, training was commenced 24 h after MCAO. However, we commenced training three days after MCAO. Previous research has reported that mobilization within 24 h after hospitalization caused negative effects on functional outcome, death rate, and dependency [32]. The exercise intensity used in our study was 7 m/min, while that in Zheng et al.’s study was 20 m/min. Moreover, Pei et al. [33] reported that 8 m/min exercise intensity significantly increased levels of BDNF and caused positive effects on the hippocampus compared to that in the control group. They also reported that 20 m/min exercise intensity increased levels of corticosterone, which implies a greater stress response. These findings suggest that low-intensity aerobic exercise may be more effective for recovery in middle cerebral ischemia rats. We used low intensity training with high frequency rTMS, which exhibited beneficial effects. We speculate that rTMS and exercise may induce synergistic effects.

Neuroplasticity, as indicated by changes in MEP amplitude and augmented BDNF/TrkB signaling proteins, suggest that rTMS combined with running exercise had a positive effect on motor function in MCAO rats. We hypothesize that combining multiple rehabilitation strategies can result in superior clinical outcomes compared to those using a single strategy. In particular, rTMS induced neuroplastic and neuroprotective effects on lesioned brains. Indeed, combined rTMS and exercise produced beneficial effects on motor recovery and neuroplastic changes in this study.

There were several limitations of our study that should be considered when interpreting our findings. The first limitation is that the experimental design of this trial did not include an exercise only group, which may clarify the synergistic effectiveness of combined treatment. Thus, we recommend that the next study should focus on the synergistic effect of aerobic exercise and brain stimulation including the exercise only group. Second, although we performed training before surgery to reduce the effect of habituation, the effect of motor skill learning could affect the rotarod results. Third, our sample sizes were small, thus the study lacked power. Fourth, our observation period did not encompass comprehensive recovery; thus, we were unable to determine changes after two weeks. Fifth, TTC staining for the infarction area at 14 days after stroke could attribute to the discrepancy with previous animal study. Finally, although we used the smallest coil, its size was relatively large for 8-week-old rats. Further studies should address these limitations.

## 6. Conclusions

In conclusion, low-intensity aerobic exercise with high-frequency rTMS has additive effects on stroke recovery including neuroplastic changes rather than in high-frequency rTMS only. These results suggest that rTMS combined with exercise therapy is a promising strategy for stroke rehabilitation and can be used as an adjunctive treatment for promoting motor function recovery in stroke patients.

## Figures and Tables

**Figure 1 brainsci-10-00186-f001:**
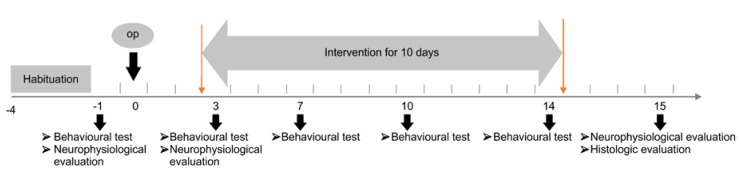
Timeline of the intervention experiment and evaluation.

**Figure 2 brainsci-10-00186-f002:**
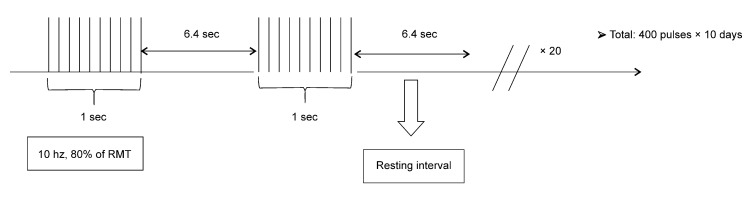
Stimulation protocol of repetitive transcranial magnetic stimulation; RMT: resting motor threshold.

**Figure 3 brainsci-10-00186-f003:**
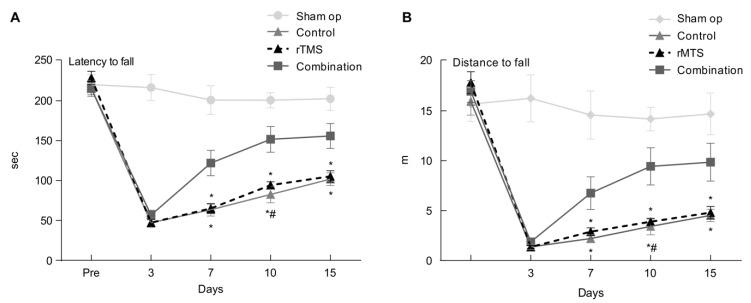
The latency to fall (**A**) and distance to fall (**B**) of the rotarod test showed a significant decrease from three days to 14 days in the permanent middle cerebral artery occlusion rat model compared with pre-operation. The combination group had a significant difference with rTMS and control groups (*p* < 0.05). Values are mean ± SEM. * *p* < 0.17 versus combination group. # *p* < 0.17 versus the rTMS group.

**Figure 4 brainsci-10-00186-f004:**
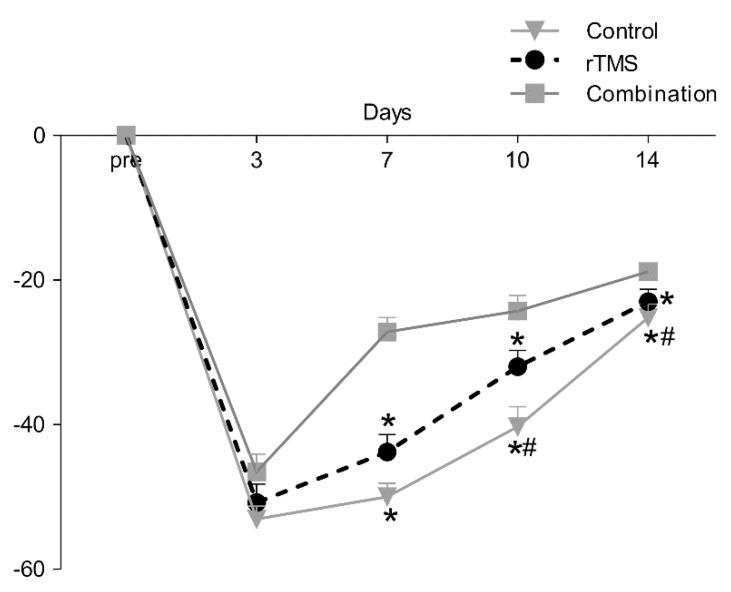
The total score for the Garcia test (% of pre) are shown. The significant decrease from three days to 14 days in the permanent middle cerebral artery occlusion rat model compared with pre-operation. The data showed that there was a significant difference among the three group (*p* < 0.05).Values are mean ± SEM. * *p* < 0.017 versus combination group. # *p* < 0.017 versus the rTMS group.

**Figure 5 brainsci-10-00186-f005:**
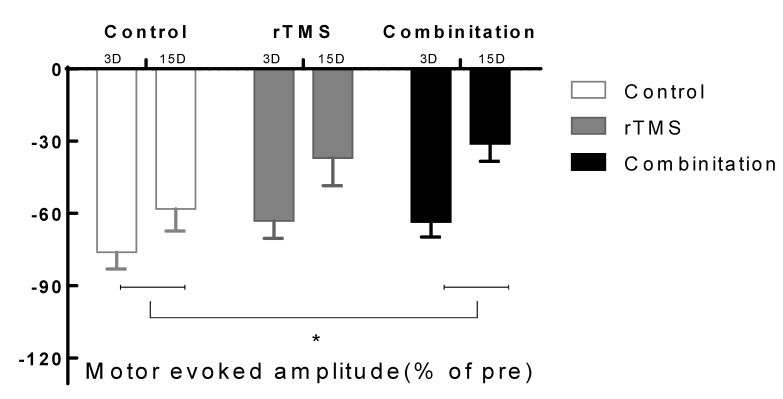
Motor evoked potentials amplitude change between 3 and 15 days. A significant difference was seen in the combination group compared with the control group (* *p* < 0.017). Values are mean ± SEM. * *p* < 0.17 versus combination group.

**Figure 6 brainsci-10-00186-f006:**
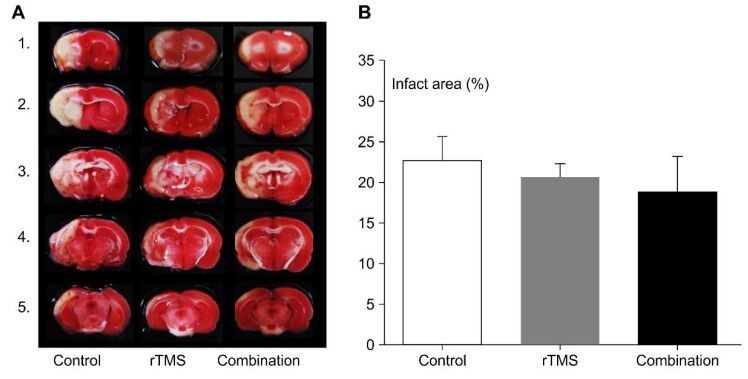
Triphenyltetrazolium chloride (TTC) stain for rat brain at 15 days (**A**), the infarction area of the three group did not show significant difference (**B**). Values are mean ± SEM.

**Figure 7 brainsci-10-00186-f007:**
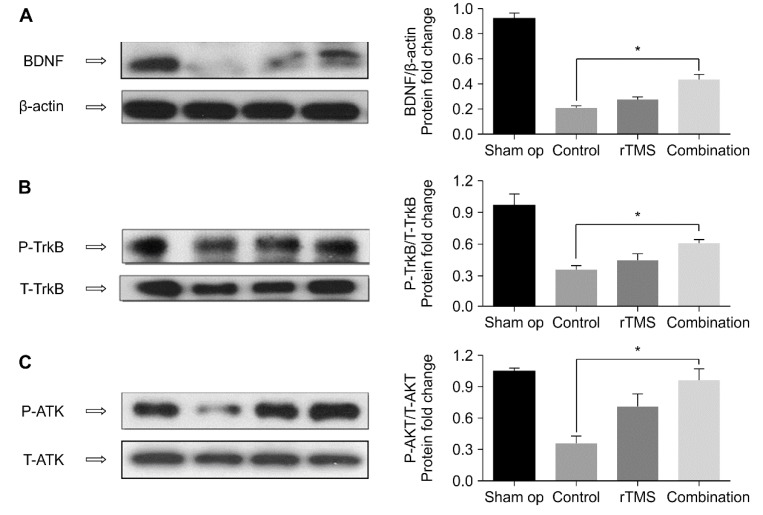
The proteins of BDNF (**A**), p-TrkB (**B**), p-AKT (**C**) in the BDNF-TrkB signaling pathways were higher in the combination group compared with the control group. * *p* < 0.017. Values are mean ± SEM.

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
