# Peer review of "Effects of Repetitive Transcranial Magnetic Stimulation (rTMS) Combined with Aerobic Exercise on the Recovery of Motor Function in Ischemic Stroke Rat Model"

_brainsci, 2020, doi:10.3390/brainsci10030186_

Round 1

Reviewer 1 Report

This manuscript examines the synergistic effects of combined neuromodulation (rTMS ) and aerobic exercise on motor recovery and neural plasticity in a rodent model of stroke.  In this study, the authors compare how rTMS, combined rTMS and exercise, sham rTMS differentiate in their recovery from experimental stroke using measures of motor behavior (rotarod test), MEP amplitude, infarct volume and expression of BDNF pathway proteins.  Research using animal models of stroke and rTMS are critical to our understanding of neuromodulation in stroke recovery as they provide us insight into the biological mechanisms underlying recovery.  The authors in this study found that measures of motor recovery, BDNF pathway proteins, and MEPs improved significantly more in the combination group than the control or rTMS only groups suggesting there may indeed by a synergistic effect of exercise and rTMS.  While the question investigated by this study would have a significant impact in our understanding of rTMS and exercise in recovery after stroke, there are some issues that should be addressed to improve the overall quality of the manuscript:

  1. Why was a suprathreshold intensity of stimulation not used for the rTMS intervention? A rationale for using a subthreshold rTMS stimulation (80% of rMT) should be discussed within the methods section.

  1. Was there a large variability in resting motor thresholds between groups? RMT values (%MSO) should be reported for each animal in the results section.

  1. How was the placement of the coil for measuring MEPs kept consistent between different days? Variations in coil position can significantly effect MEP amplitude.

  1. Units are missing in the results section for multiple measures including latency to fall, and MEP amplitude.

  1. There are a number of grammatical errors that should be checked. The quality of writing can be improved from a sentence structure standpoint to increase readability and clarity throughout.

  1. Is there any chance peripheral nerves or spinal cord were stimulated by the figure-of-eight coil? How large of a portion of each animal’s cortex was stimulated by rTMS and single pulse TMS?  This can be estimated by determining the size of the field produced by the TMS coil.   

  1. It would be interesting to know if the magnitude of change in BDNF, MEPs and behavior from of exercise alone. Although the authors acknowledge this in their discussion/ limitation section, I suggest the authors try to compare the effect size of their results to previous literature when exercise alone is performed. 

  1. Seeing that these they both associated with neuroplasticity would be interesting to see if there is a correlation between increases in BDNF signaling and change in MEP amplitude.

  1. How do we know that the effects seen by the exercise group were not simply due to habituation or motor skill learning due to time spent on the rotarod? This should be discussed more as a limitation.    

Author Response

Dear reviewer:

We would like to thank you for taking the time to review our article. We have made some corrections and clarifications in the manuscript after going over your comments.

The changes are summarized below:

Comment:

1. Why was a suprathreshold intensity of stimulation not used for the rTMS intervention? A rationale for using a subthreshold rTMS stimulation (80% of rMT) should be discussed within the methods section.

Our response:  

  1. Luo et al. [1] used 20 Hz. 80% RMT. Kim et al. [2] used 10 Hz, 80% RMT reported that this improved cortex excitability. Yoon et al. [3] used 10hz, 80% RMT. We have added the reference in the Methods section. (page 3, line 117).
  2. RMT is to induce the intensity of 50uv. High-intensity stimulation of the motor cortex will be accompanied by the reflection of the limbs. Adding 400 pulses to the stimulation may cause discomfort.
  3. In addition, considering the safety guideline for using rTMS [4], side effects of high-frequency rTMS is epilepsy
    1. Highfrequency Repetitive Transcranial Magnetic Stimulation (rTMS) Improves Functional Recovery by Enhancing Neurogenesis and Activating BDNF/TrkB Signaling in Ischemic Rats.
    2. Evidence-based guidelines on the therapeutic use of repetitive transcranial magnetic stimulation (rTMS)
    3. Mechanism of functional recovery after repetitive transcranial magnetic stimulation (rTMS) in the subacute cerebral ischemic rat model: neural plasticity or anti-apoptosis?
    4. Safety, ethical considerations, and application guidelines for the use of transcranial magnetic stimulation in clinical practice and research.

2. Was there a large variability in resting motor thresholds between groups? RMT values (%MSO) should be reported for each animal in the results section.

Our response: We found the intensity of RMT on the right hemisphere before the MCAO procedure. We conducted a pilot trial to determine RMT for forepaw in 10 healthy rats before the main experiment. The pilot trial showed that average RMT for biceps muscle was 37% of machine output. We have added the part in the Methods section. (page 3, line 113).

3. How was the placement of the coil for measuring MEPs kept consistent between different days? Variations in coil position can significantly effect MEP amplitude.

Our response: Hot spot was measured and marked before surgery, we marked with a pen at a specified point, and we tried our best to locate the specified point every time during treatment and test. (page 3, line 111).

4. Units are missing in the results section for multiple measures including latency to fall, and MEP amplitude.

Our response: We highly appreciate your guidance, we have added.

5. There are a number of grammatical errors that should be checked. The quality of writing can be improved from a sentence structure standpoint to increase readability and clarity throughout.

Our response: We highly appreciate your comments. We have corrected the paper.

6. Is there any chance peripheral nerves or spinal cord were stimulated by the figure-of-eight coil? How large of a portion of each animal’s cortex was stimulated by rTMS and single pulse TMS? This can be estimated by determining the size of the field produced by the TMS coil.

Our response:

  1. Figure-of-eight coil (cool-B35; 24 mm inner diameter, 47 mm outer diameter.) We have added the size in the Methods section (page 3, line 107).
  2. When stimulating, we observed lateralized TMS produced a twitch only in the contralateral forepaw and shoulder.

7. It would be interesting to know if the magnitude of change in BDNF, MEPs and behavior from of exercise alone. Although the authors acknowledge this in their discussion/ limitation section, I suggest the authors try to compare the effect size of their results to previous literature when exercise alone is performed.

Our response: We highly appreciate your suggestion. We looked for many papers on MCAO rats for exercise, but unfortunately,we cannot find papers that check BDNF, MEPs, and behavior at similar points in time and reveal Mean and SD values.

8. Seeing that these they both associated with neuroplasticity would be interesting to see if there is a correlation between increases in BDNF signaling and change in MEP amplitude.

Our response: We focused on comparing the beneficial effect on the behavioral recovery of rTMS only to rTMS + aerobic exercise. We analyzed the correlation between BDNF and MEP and found a significant positive relationship between them. There is a disparity between our main goal and this correlation analysis. So, we could add the result in the appendix.

(r = 0.598,  p<0.05)

9. How do we know that the effects seen by the exercise group were not simply due to habituation or motor skill learning due to time spent on the rotarod? This should be discussed more as a limitation.

Our response: All rats were habituated(page 2, line 75)on the rotarod equipment for three days, before the pre-test. Therefore, we think it is not necessary to consider the effect of habituation. As your comment, the rotarod result included the effect of motor skill learning. We add this part to the limitation. (page10, line 360).

Reviewer 2 Report

In this study, the authors investigated the therapeutic effects of repetitive transcranial magnetic stimulation (rTMS) alone or combined with the aerobic exercise on neuroplasticity and recovery of motor function in a rat model of permanent middle cerebral artery occlusion (pMCAO). The high-frequency rTMS(10Hz) therapy on the ipsilesional forepaw motor cortex started at 3 days after MCAO and was administered daily for 10 secessions. In combination group, the aerobic exercise training on rotarod was followed rTMS therapy daily. At 15 days post-pMCAO, the combination therapy significantly improved neurological deficits and motor evoked amplitude. These benefits were associated with higher protein levels of BDNF, p-TrkB and p-AKT in the brain tissues. Low intensity aerobic exercise with high frequency rTMS showed the synergistic beneficial effects on stroke, which can be used as a treatment for recovery of the motor function in stroke patients.

This is a descriptive study. The specific comments are as follows:

  1. Previous studies have shown the benefits of aerobic exercise alone in rehabilitation after stroke. It is necessary to include exercise only group to draw the conclusion about synergistic effects of rTMS combined with exercise.
  2. The authors’ own published work has demonstrated that similar rTMS (800 stimulation /session) therapeutic regimen improved the rotarod performance and amplitude of MEP in rat model of pMCAO (Jin, 2018). In this study, however, rTMS was applied at 400 stimulation /session) that resulted relative less efficacy of protection regarding to results of rotarod and MEP of rTMS only group. Why the authors use less intensity of stimulation in this study?
  3. The author interpreted the results that the difference between two behavioral tests could reflect the repetitive exercise effect. If the rats in combined therapy group used rotarod every day, would they potentially perform much better due to the training effects? However, the result showed the opposite direction. Again, aerobic exercise alone group will help clarify this question.
  4. The role of Akt in BDNF/TrkB signaling need to be discussed briefly.
  5. WB was performed in the peri-infarct region of ipsilateral brain cortex. How were this region determined and dissected from infarction area?
  6. Figures

-  Figure 3:  The sizes of two graphs are not consistent.

-   Figure 6 A: TTC is not an appropriate method to evaluate infarct size at 15 days after MCAO.  

  1. The manuscript needs the proof reading. There are some English spelling and formatting errors in the manuscript.

Author Response

Dear reviewer:

We would like thank you for taking the time to review our article. We have made some corrections and clarifications in the manuscript after going over your comments.

The changes are summarized below:

Comment:

1. Previous studies have shown the benefits of aerobic exercise alone in rehabilitation after stroke. It is necessary to include exercise only group to draw the conclusion about synergistic effects of rTMS combined with exercise.

Our response: As your comment, we regret that the study design for this study was not perfect. We will make a plan with exercise only group for next experiments.

2. The authors’ own published work has demonstrated that similar rTMS (800 stimulation /session) therapeutic regimen improved the rotarod performance and amplitude of MEP in rat model of pMCAO (Jin, 2018). In this study, however, rTMS was applied at 400 stimulation /session) that resulted relative less efficacy of protection regarding to results of rotarod and MEP of rTMS only group. Why the authors use less intensity of stimulation in this study?

Our response: After rTMS 800 pulse stimulation, we found that it was hard to walk in the subsequent exercise therapy. So we referred to the rTMS protocol of other papers [1]. We have concerns about the safety of the amount and duration of treatment after MCAO.

  1. Mechanism of functional recovery after repetitive transcranial magnetic stimulation (rTMS) in the subacute cerebral ischemic rat model: neural plasticity or anti-apoptosis? (350 stimulation/session)

3. The author interpreted the results that the difference between two behavioral tests could reflect the repetitive exercise effect. If the rats in combined therapy group used rotarod every day, would they potentially perform much better due to the training effects? However, the result showed the opposite direction. Again, aerobic exercise alone group will help clarify this question.

Our response: In rotarod result, although all rats were habituated on the rotarod equipment for three days, before the pre-test, as your comment, the rotarod results will include the effect of motor skills learning. We add this section to the limitation. Garcia's test results complement the rotarod results to some extent.

4. The role of Akt in BDNF/TrkB signaling need to be discussed briefly.

Our response: We highly appreciate your suggestion. AKT is one of the downstream signaling pathways to BDNF activates TrkB. We have added the reference in the introduction section (page 2, line 55).

5. WB was performed in the peri-infarct region of ipsilateral brain cortex. How were this region determined and dissected from infarction area?

Our response: We cut the brain into 5 slices with same method of TTC. We collect the tissue from the third slice. Under the light, we can observe that the color of the lesion area is different from that of normal tissues, and the color of the diseased area will appear whiter than normal area. Due to the 15 days after stroke, the marginal tissue appears loose and coarse. Depending on the color and tissue density change, we could find the marginal line and then get the peri-infarct tissue along the marginal line.

6. Figures

-  Figure 3:  The sizes of two graphs are not consistent.

Our response:

Figure 3: We have modified.

-  Figure 6 A: TTC is not an appropriate method to evaluate infarct size at 15 days after MCAO. 

Our response:

Figure 6 A: We have referred to some papers [1,2,3], and then decided this time point. It is estimated that our understanding of the time point of TTC measurement of infarct area is not enough. Thank you for your comments. We will pay attention to the time point of TTC in future experiments.

  1. Effect of willed movement training on neurorehabilitation after focal cerebral ischemia and on the neural plasticity-associated signaling pathway (5 days)
  2. Treadmill exercise improves neurological function by inhibiting autophagy and the binding of HMGB1 to Beclin1 in MCAO juvenile rats (7days)
  3. High-Frequency Repetitive Transcranial Magnetic Stimulation (rTMS) Improves Functional Recovery by Enhancing Neurogenesis and Activating BDNF/TrkB Signaling in Ischemic Rats (Nissl staining: 14days)

7. The manuscript needs the proof reading. There are some English spelling and formatting errors in the manuscript.

Our response: We highly appreciate your comments. We have corrected the paper.

Round 2

Reviewer 1 Report

I would like to thank the authors for their edits.  I believe it reads better now and most of the major concerns have been addressed. Some minor comments remain:

  1. The approximated magnetic field on the brain should be discussed.  This is different from the size of the TMS coil.  This can be estimated using the strength of the magnetic field and distance of the coil from the brain. 
  2. The grammar in the final sentence in the introduction describing the hypothesis is still incorrect and the sentence is not very clear.  This should be corrected to make the objective of the study clear.     

Author Response

Dear reviewer:

We would like thank you and the reviewers of Scientific Reports for taking the time to review our article. We have made some corrections and clarifications in the manuscript after going over the reviewers’ comments.

The changes are summarized below:

Comment:

1. The approximated magnetic field on the brain should be discussed.  This is different from the size of the TMS coil.  This can be estimated using the strength of the magnetic field and distance of the coil from the brain.

Our response: A magnetic stimulator (MagPro X100, MagVenture, Inc., Alpharetta, GA, USA) with a figure-of-eight coil (cool-B35; 24 mm inner diameter, 47 mm outer diameter) was used to stimulate conscious rats 3 days after cerebral ischemia. The maximal initial dB/dt near the coil surface was 50 kT/s and active pulse width (biphasic) was 290 μs according to the magnetic and electrical properties sheet of cool-B35 coil supplied by manufacture. (page 3, line 109)

2. The grammar in the final sentence in the introduction describing the hypothesis is still incorrect and the sentence is not very clear.  This should be corrected to make the objective of the study clear. 

Our response: We highly appreciate your comments, we modified.We hypothesis that low-intensity aerobic exercise with high-frequency rTMS could induce better motor recovery and make neuroplastic change after stroke than rTMS only. (page 2, line 67)

Reviewer 2 Report

The authors addressed most of the questions/comments in the revision. The concern study of study design and method of TTC staining are limitations of current study that require validation in future study. Thus “the synergistic beneficial effects” in Conclusion need to be rephrased.

Author Response

Dear reviewer:

We would like thank you and the reviewers of Scientific Reports for taking the time to review our article. We have made some corrections and clarifications in the manuscript after going over the reviewers’ comments.

The changes are summarized below:

Comment:

The authors addressed most of the questions/comments in the revision. The concern study design and method of TTC staining are limitations of current study that require validation in future study. Thus “the synergistic beneficial effects” in Conclusion need to be rephrased.

Our response: We discuss about using TTC stain at 14 days after stroke. (page 9, line 329) And for the study design, we rephrase the conclusion as “In conclusion, low-intensity aerobic exercise with high-frequency rTMS have additive effects on stroke recovery including neuroplastic changes rather than high-frequency rTMS only. These results suggest that rTMS combined with exercise therapy is a promising strategy for stroke rehabilitation and can be used as an adjunctive treatment for promoting motor function recovery in stroke patients.” (page 10, line 368)